# Revalorization of Coffee Husk: Modeling and Optimizing the Green Sustainable Extraction of Phenolic Compounds

**DOI:** 10.3390/foods10030653

**Published:** 2021-03-19

**Authors:** Miguel Rebollo-Hernanz, Silvia Cañas, Diego Taladrid, Vanesa Benítez, Begoña Bartolomé, Yolanda Aguilera, María A. Martín-Cabrejas

**Affiliations:** 1Department of Agricultural Chemistry and Food Science, Universidad Autónoma de Madrid, 28049 Madrid, Spain; miguel.rebollo@uam.es (M.R.-H.); silvia.cannas@uam.es (S.C.); vanesa.benitez@uam.es (V.B.); maria.martin@uam.es (M.A.M.-C.); 2Institute of Food Science Research, CIAL (UAM-CSIC), 28049 Madrid, Spain; d.taladrid@csic.es (D.T.); b.bartolome@csic.es (B.B.)

**Keywords:** coffee by-products, phenolic compounds, antioxidant capacity, response surface methodology, artificial neural networks

## Abstract

This study aimed to model and optimize a green sustainable extraction method of phenolic compounds from the coffee husk. Response surface methodology (RSM) and artificial neural networks (ANNs) were used to model the impact of extraction variables (temperature, time, acidity, and solid-to-liquid ratio) on the recovery of phenolic compounds. All responses were fitted to the RSM and ANN model, which revealed high estimation capabilities. The main factors affecting phenolic extraction were temperature, followed by solid-to-liquid ratio, and acidity. The optimal extraction conditions were 100 °C, 90 min, 0% citric acid, and 0.02 g coffee husk mL^−1^. Under these conditions, experimental values for total phenolic compounds, flavonoids, flavanols, proanthocyanidins, phenolic acids, *o*-diphenols, and in vitro antioxidant capacity matched with predicted ones, therefore, validating the model. The presence of chlorogenic, protocatechuic, caffeic, and gallic acids and kaemferol-3-*O*-galactoside was confirmed by UPLC-ESI-MS/MS. The phenolic aqueous extracts from the coffee husk could be used as sustainable food ingredients and nutraceutical products.

## 1. Introduction

The United Nations Sustainable Development Goals promote the warranting of sustainable consumption and production patterns such as achieving the efficient use of natural resources and reducing waste generation through prevention, reduction, recycling, and reusing [1]. Likewise, the Food and Agriculture Organization (FAO) is focused on redesigning the global food system to be more productive, environmentally sustainable, and able to deliver healthy and nutritious foods to the population [2]. In this respect, the reduction of the food industry wastes and by-products is a clear need. The development of ground-breaking strategies for revalorizing food by-products via their conversion into food-grade novel ingredients is a critical challenge in the food chain [3].

The processing of the coffee cherries into coffee beans is very complex and produces a variety of residues. Mature fruits are collected and transformed by basically two methods: dry or wet processing [4]. Within the dry processing, the coffee cherries are sun-dried. The coffee husk is separated during the following de-hulling step. The coffee husk comprises the skin, pulp, mucilage, parchment, and parts of the silverskin. In the wet processing, conversely, water is used to separate ripened and unripped coffee cherries. After this separation, the coffee pulp and skin are removed using a pulper. The coffee parchment remains attached to the coffee bean [5]. Therefore, wet processing produces pulp and parchment, whereas dry processing produces just the coffee husk, which contains both parts (Figure 1). The disposal of coffee husk without treatment causes environmental problems in the producing countries, owing to the high content of caffeine and phenolic compounds composing it [6]. The traditional use of the coffee husk is the preparation of the “Cascara beverage”, traditionally consumed in Yemen and Ethiopia [7]. Current uses for the coffee husk include mushroom cultivation [8], lignin extraction [9], and recovery of biomolecules from the lignin alkali hydrolysate [10], bioethanol production [11], bio-sorbents preparation [12], and composting [13]. Although these strategies for valorizing the coffee husk contribute to the coffee industry’s sustainability, the production of high-value-added food ingredients is a more desirable approach to promoting a more productive, environmentally sustainable food system.

The coffee husk is a source of phenolic compounds and caffeine. Chlorogenic, protocatechuic, and gallic acids are the main phenolic compounds comprised in it and responsible for the coffee husk antioxidant properties [7]. Extracting these compounds could be a strategy to valorize the coffee husk and develop food ingredients with health-promoting properties. Coffee antioxidants are associated with preventing chronic diseases such as obesity, diabetes, and other cardiometabolic diseases [14]. Limited research has evaluated the extraction of these phytochemicals from the coffee husk [15]. Green extraction methods are needed to ensure the sustainability of the system. Even though there is research pointing out the use of ultrasound- and microwave-assisted extractions and pressurized liquid extraction [16], heat-assisted extraction (HAE) is still the most common extraction technique in the food industry [17]. Consequently, there is a need to model, optimize, and validate the green HAE of phenolic compounds from the coffee husk.

Response surface methodology (RSM) is one of the chemometric techniques often employed to optimize procedures in food production and analysis [18]. The parameters influencing the extraction are modeled and optimized using RSM, principally to reduce energy and, therefore, extraction costs [19]. Machine learning algorithms may also be employed for these purposes. Artificial neural networks (ANNs) have gained interest in modeling and optimization processes. This methodology allows for the study of the relationships between the extraction and response variables, i.e., the extraction yield, employing fewer experimental measurements [20]. ANNs are computational models based on the structure and functions of the nervous system and the brain and have extraordinary learning and predictive abilities [21]. Hence, these mathematical and computational tools can be used to enhance the effectiveness of the extraction process, making it more economically and environmentally sustainable.

We hypothesized that the modification of extraction parameters would increase the recovery yield of phenolic compounds from the coffee husk, allowing the establishment of a green sustainable extraction method. Hence, this study aimed to model the process conditions to maximize the sustainable aqueous extraction of phenolic compounds from the coffee husk using response surface methodology and artificial neural networks, optimize it, and comprehensively characterize the obtained extracts using UPLC-MS/MS analysis. Multivariate statistics were used to gain insight into the effects of extraction conditions on the phenolic composition and its relationship with the in vitro antioxidant capacity of the obtained aqueous phenolic extracts.

## 2. Materials and Methods

### 2.1. Material and Sample Preparation

The coffee husk, mechanically separated from the sun-dried cherries of the Arabica species variety Caturra, was supplied by “Las Morenitas” (Nicaragua). The milling of the coffee husk was carried out in a pilot-scale ball mill over three days, then sieved with a pilot-scale sieve, selecting the fraction with a particle size of <250 μm. Milled coffee husk was stored in closed and sealed plastic bags and preserved in dark and dry conditions to avoid oxidation until further extraction and analysis.

### 2.2. Experimental Design

#### 2.2.1. Response Surface Methodology (RSM)

Box–Behnken, being a spherical RSM, consists of a central point and several middle points on the edges of a cube superimposed on the sphere, which requires fewer experiments than other statistical designs. We employed a four-factor, three-level Box–Behnken design coupled to RSM to find the optimal extraction conditions to achieve the highest extraction of phenolic compounds from the coffee husk. The experimental conditions for the aqueous extraction of phytochemicals from the coffee husk are presented in Table 1. The statistical design comprised 27 experimental runs with three levels (−1, 0, 1) for each of the variables: temperature (°C) (*X_1_*), time (min) (*X_2_*), acidity as the percentage of citric acid in water (%) (*X_3_*), and solid-to-liquid ratio (S/L, g mL^−1^) (*X_4_*). Those parameters were selected according to previous studies found in the literature and tested on preliminary experiments to guarantee they exerted an influence on the extraction of phenolic compounds from coffee parchment [17]. The impact of extraction temperature was investigated in the range from 30 to 100 °C, time from 5 to 90 min, S/L ratio, 0.02–0.05 g mL^−1^, and acidity, 0–2% citric acid. The variables were coded according to the following equation:(1)$$X=\frac{x_{i}-\;x_{0}}{\Delta x}$$
where *X* is the coded value; *x_i_*, the corresponding actual value; *x_0_*, the real value at the center of the domain; and *Δx*, the increment of *x_i_* corresponding to a variation of 1 unit of *x*. The response variables were fitted to the following second-order polynomial model equation, which described the relationship between the responses and the independent variables.
(2)$$Y=\;\beta_{0}+\;\overset{k}{\underset{i=1}{\sum }}\beta_{i}X_{i}\;+\overset{k}{\underset{i=1}{\sum }}\beta_{ii}X_{ii}^{2}+\overset{k-1}{\underset{i}{\sum }}\overset{k}{\underset{j}{\sum }}\beta_{ij}X_{ij}\;$$
where *Y* was the response variables; *X_i_* and *X_j_* were independent coded variables; *β_0_* was the constant coefficient; *β_i_* was the linear coefficient; *β_ii_* was the quadratic coefficient, and *β_ij_* was the cross-product coefficients.

Based on the analysis of variance (ANOVA), the regression coefficients of individual linear, interaction, and quadratic terms were determined. The numerical magnitude of the standardized model coefficients evidenced their significance in the obtained model. Among standardized coefficients, the larger values are more effective. Plots depicting response surface 3D plots were constructed for all the response variables (Figure 1A). 

The polynomial equation’s fitness to the responses was assessed through the coefficient of determination (*R^2^*). The significance of all the terms within the polynomial equation was analyzed statistically by analyzing the *F*-value at *p* < 0.05. Equations were created, selecting the significant (*p* < 0.05) non-standardized coefficients (including non-significant terms if needed to ensure that the model was hierarchical), and their statistical parameters (*F*-value and *R^2^*) were determined again.

#### 2.2.2. Artificial Neural Networks (ANNs)

A multilayer perceptron (MLP)-based feed-forward ANN was applied for modeling the extraction of phenolic compounds from the coffee husk. MATLAB version R2020a was used to model the data using ANNs. The experimental data was constructed using the regression-based network approach. The Broyden–Fletcher–Goldfarb–Shanno (BFGS) quasi-Newton back-propagation (TRAINBFG) method was selected since it is an efficient training function because it performs non-smooth optimizations and smaller networks [22]. The gradient descent method (LEARNGDM) as the adaptive learning function was used to minimize the mean squared error (MSE) between the network output and the actual error rate [23]. The hyperbolic tangent sigmoid transfer function (TANSIG) and linear transfer function (PURELIN) were used to calculate a layer’s output from its net input [24]. All these functions were used to train the neural network and built the best ANN. Multiple feed-forward neural networks were trained and, subsequently, tested by determining the number of neurons in the hidden layer to select an optimized ANN topology, with the lowest root mean square error (RMSE) and highest *R^2^* values. However, the number of epochs (or cycles through the whole training dataset) was restricted to a minimum to avoid over-fitting while establishing an optimal topology. Increased epoch numbers may cause model over-fitting issues. The network architecture (Figure 2B) consisted of an input layer with four neurons (temperature (T), time, (t), acidity, and S/L ratio), one hidden layer with ten neurons, and an output layer with one neuron, which represented each of the response variables (total phenolic compounds, TPC, in Figure 2B).

### 2.3. Comparison of the Prediction Ability of RSM and ANN

Several statistical parameters, including the coefficient of determination (*R^2^*), the root mean square error (RMSE), and the absolute average deviation (AAD), were calculated for the comparison of the estimation capabilities of RSM and ANN, according to the following equations.
(3)$$R^{2}=1-\;\frac{\sum_{i=1}^{n}{\left(Y_{pre}-\;Y_{exp}\right)}^{2}}{\sum_{i=1}^{n}{\left(Y_{m}-\;Y_{exp}\right)}^{2}}$$
(4)$$\mathrm{RMSE}=\;\sqrt{\frac{\sum_{i=1}^{n}{\left(Y_{pre}-\;Y_{exp}\right)}^{2}}{n}}$$
(5)$$\mathrm{ADD}\;\left(\%\right)=\;\left(\frac{\sum_{i=1}^{n}{\left(\left|Y_{pre}-\;Y_{exp}\right|/Y_{pre}\right)}^{2}}{n}\right)\cdot 100$$
where *Y_pre_* is the predicted response variable (by either RSM or ANN), *Y_exp_* is the observed response variable, *Y_m_* is the average response variable, and *n* is the number of experiments.

### 2.4. Validation of the Model

The extraction conditions were optimized for the maximum yield of phenolic compounds (total phenolic compounds (TPC), total flavonoids (TF), total flavanols (TFL), total proanthocyanidins (PAC), total phenolic acid (TPA), and total *ortho*-diphenols (TOD)) and the antioxidant capacity (AC) by employing RSM. Then, the responses were determined under the optimal and suboptimal extraction conditions. Finally, the experimental values were compared with predicted values (from RSM and ANN) based on the coefficient of variation, CV (%), to determine the model’s validity. The UPLC-ESI-MS/MS profiles of phenolic compounds were also determined at the optimized conditions.

### 2.5. Heat-Assisted Extraction (HAE)

The HAE was carried out in closed vessels in a temperature-controlled water bath with continuous stirring. According to the experimental design, the milled coffee husk was extracted at various temperatures, times, acidity values, and S/L ratios as described in Table 1. Once HAE was finished, the solubilized phytochemicals were separated by centrifugation (4000× *g*, 4 °C, 15 min), and the supernatants were freeze-dried. The samples were resuspended in Milli-Q water (10 mL) after neutralization and preserved at −20 °C until analysis.

### 2.6. Organic Solvent Extraction of Free and Bound Phenolic Compound Fractions

Free and bound phenolic fractions from the coffee husk were extracted as described by Rebollo-Hernanz et al. [25] Phenolic compounds from the coffee husk were recovered using a conventional organic method to compare these conditions to those of the optimized methodology using just water as the extracting agent. Here, phenolic compounds’ total content was calculated as the sum of the free and bound phenolic fraction.

#### 2.6.1. Extraction of Free Phenolic Compounds

Milled coffee husk (1.0 g) was macerated for 30 min using methanol-HCl (0.1%)/H_2_O (80:20, *v*/*v*) (50 mL) in an ultrasonic bath. After that, samples were maintained under stirring for 16 h at 40 °C. The samples were centrifuged (4000× *g*, 4 °C, 15 min), and the supernatants were collected. This process was repeated two times. All the methanolic fractions were combined and evaporated under vacuum. The free phenolic compound fractions were redissolved in methanol (10 mL) and were preserved at −20 °C until analysis.

#### 2.6.2. Extraction of Bound Phenolic Compounds

The insoluble residues from the free phenolic compound extraction were hydrolyzed using 4 mol L^−1^ NaOH (20 mL) under an atmosphere of N_2_ under continuous shaking (1 h, 25 °C). The hydrolysates were acidified with 8 mol L^−1^ HCl until reaching pH 2. Then, the samples were centrifuged (4000× *g*, 4 °C, 15 min), and bound phenolic compounds were extracted from the aqueous alkali phase. Liquid:liquid extraction with diethyl ether:ethyl acetate (50:50, *v*/*v*) was repeated three times, and the three organic phases were mixed. Organic fractions were dried under vacuum, redissolved in methanol (10 mL), and preserved at −20 °C until analysis.

### 2.7. Determination of Phenolic Compounds

#### 2.7.1. Total Phenolic Compounds (TPC)

Total phenolic compounds were quantified using the Folin–Ciocalteu colorimetric method, following the protocol of Singleton, Orthofer, and Lamuela-Raventós [26] adapted to the micromethod format. Samples (10 µL) were mixed with the Folin–Ciocalteu reagent (diluted 1:14, *v*/*v* in Milli-Q water) (150 µL). After incubating for 3 min, 20% Na_2_CO_3_ (50 μL) was added to each well, and the mixture was homogenized. Plates were incubated for 2 h at room temperature. The absorbance was read at 750 nm using a microplate reader incubation. Calibration curves were prepared using solutions of gallic acid, and results were expressed as mg of gallic acid equivalents per gram (mg GAE g^−1^) of dry coffee husk.

#### 2.7.2. Total Flavonoids (TF)

The content of total flavonoids was determined using the aluminum chloride method adjusted to the micromethod format [27]. Briefly, samples and standards (100 µL) were mixed with 5% Na_2_NO_2_ (30 µL) and incubated for 5 min at 20 °C. Subsequently, 10% AlCl_3_ (30 µL) was added. The mixture was further homogenized and incubated for 6 min. Then, 2 mol L^−1^ NaOH (100 µL) was added, and the solution was finally homogenized. The absorbance was recorded at 510 nm. The content of total flavonoids was estimated with a quercetin calibration curve, and the results were expressed as mg of quercetin equivalents per gram (mg QE g^−1^) of dry coffee husk.

#### 2.7.3. Total Flavanols (TFL)

The content of total flavanols was assessed by the vanillin method adapted [28]. Samples (10 µL) were added to each well, and 8.4 mol L^−1^ vanillin 1% HCl (50 μL) and 37% HCl (250 μL) were added and let to react (15 min, 20 °C). The absorbance was measured at 500 nm, and the concentration of total flavanols was calculated using a standard curve of catechin. The results were expressed as mg of catechin equivalents per gram (mg CE g^−1^) of dry coffee husk.

#### 2.7.4. Total Proanthocyanidins (PAC)

The content of total proanthocyanidin was determined using a modification of the Bate-Smith method [29]. Briefly, 10 µL of each extract and 1 mL of 0.54 mmol L^−1^ FeSO_4_ in butanol/HCl (50:50) were incubated at 90 °C for 1 h. After cooling, the absorbance was measured at 550 nm against an unheated blank prepared in the same way than each sample. Cyanidin chloride was used as a standard to construct the calibration curve. Results were expressed as mg of cyanidin chloride equivalents per gram of dry coffee husk (mg CCE g^−1^).

#### 2.7.5. Total Phenolic Acids (TPA)

The content of total phenolic acids was measured following the method described by Vukic et al. [30]. Samples or standards (10 μL) were mixed with Milli-Q water (50 μL). Then, a Na_2_MoO_4_ solution and 0.1 mol L^−1^ HCl (50 μL) were combined with the diluted sample, and then NaOH was added (100 μL, 0.1 mol L^−1^). The absorbance was measured at 490 nm and the content of total phenolic acids was estimated using a calibration curve of caffeic acid. The results were expressed as mg of caffeic acid equivalents per gram (mg CAE g^−1^) of dry coffee husk.

#### 2.7.6. Total ortho-Diphenols (TOD)

The content of *ortho*-diphenols was estimated according to the method described by Granato et al. [31]. Samples (50 μL) were mixed with 0.05 g mL^−1^ Na_2_MoO_4_·2H_2_O. The absorbance was measured at 370 nm after a 25 min incubation, and the *o*-diphenols content was calculated using a calibration curve of caffeic acid. The results were expressed as mg of caffeic acid equivalents per gram (mg CAE g^−1^) of dry coffee husk.

#### 2.7.7. Assessment of In Vitro Antioxidant Capacity (AC)

The coffee husk phenolic extracts’ in vitro antioxidant capacity was assessed using the ABTS^•+^ assay, as previously described [32]. ABTS^•+^ radical cations were generated by mixing ABTS solution with K_2_S_2_O_8_ (dark and room temperature for 12–16 h, under continuous shaking before use). The ABTS^•+^ assay solution was prepared by dilution in PBS (5 mmol L^−1^, pH 7.4) to reach an absorbance of 0.70 ± 0.02 at 734 nm. The samples and standards (30 μL) were mixed with the 270 μL of ABTS^•+^ assay solution, and the absorbance of the samples at 734 nm was read after a 10 min incubation. Calibration curves were prepared using standard solutions of Trolox, and the results were expressed as mg Trolox equivalents per gram (mg TE g^−1^) of dry coffee husk.

### 2.8. UPLC-ESI-MS/MS Analysis of Phenolic Compounds

The targeted phenolic compounds were analyzed using UPLC-ESI–MS/MS according to a method previously described [33]. Extracts were suspended in water, filtered (0.22 μm), and the internal standard 4-hydroxybenzoic-2,3,5,6-d4 acid solution (Sigma–Aldrich, St. Louis, MO, USA) was added to the samples in a proportion of 1:5 (*v*/*v*). Data were collected under the multiple reaction monitoring mode for the quantification, tracking the specific transition of parent and product ions for each compound. The electrospray ionization was operated in negative mode. All phenolics were quantified using the calibration curves of their corresponding standards. Injections were carried out in triplicate (*n* = 3).

### 2.9. Statistical Analysis

Statistical analysis of the experimental results was performed using the statistical programs Design Expert 11, MATLAB version R2020a, and SPSS 24.0. All data are presented as the mean ± standard deviation (SD) of at least three independent experiments (*n* = 3), where each experiment had a minimum of three replicates for each sample. For comparisons among extraction conditions, data were analyzed by one-way analysis of variance (ANOVA) and the post hoc Tukey test. Differences were considered significant at *p* < 0.05. The statistical design, RSM model, and optimization were calculated with Design Expert. ANN models were constructed, tested, and validated using MATLAB. The chemometric analysis was carried out to describe the phenolic extracts better. Principal component analysis (PCA) was used to classify samples according to their phenolic composition. Partial least squares analysis (PLSA) was used to rank the spectrophotometric and chromatographic parameters according to their importance (variable importance in projection (VIP) scores) on the variability among extracts. An agglomerative hierarchical cluster analysis coupled to a heatmap was generated to depict the variability among extracts. Principal components regression (PCR) and principal least squares regression (PLS-R) were constructed to evaluate the influence of individual phenolic compounds on the in vitro antioxidant capacity. Pearson’s linear correlations were calculated to assess the association between spectrophotometric techniques and results and chromatographic methods, using the concentration of phenolic compounds obtained.

## 3. Results and Discussion

### 3.1. Fitting of the RSM and ANN Models

The experimental results for each of the 27 conditions from the Box–Behnken design are presented in Table 1. The RSM fitting for each response variable (TPC, TF, TFL, PAC, TPA, TOD, and AC) was produced using second-order polynomial equations. Response surface 3D plots were generated for each response variable. Figure 2A depicts the behavior of TPC when modifying extraction variables as a representative response. The graphs were generated by plotting the response (TPC) against two independent variables while keeping the other independent variables at a fixed level (in its intermediate value). 

The non-significant terms (*p* > 0.05) were not considered to improve the models’ fitting and prediction. Both complete (RSM) and simplified models, including just significant (*p* < 0.05) terms (RSM _ST_), statistical parameters measuring the predictive ability of models are presented in Table 2. RSM models exhibited *R^2^* values between 0.8919 and 0.9744, demonstrating a high linear correlation between experimental and predicted values. The RSM _ST_ models exhibited lower *R^2^* values (0.7101–0.9597); nonetheless, the model’s mathematical fitting continues to show a strong correlation between experimental–predicted values. The lower RMSE and ADD are, the better is the fit between experimental and predicted values. Thus, it was observed that the complete RSM models exhibited lower RMSE and ADD than the RSM _ST_ models for all the response variables.

The ANN was used to predict non-linear associations between the extraction parameters (*X_1_*, *X_2_*, *X_3_*, and *X_4_*) and the response variables (TPC, TF, TFL, PAC, TPA, TOD, and AC). The experimental values used to create the RSM model were also employed to build the ANN model: 70% (19 points) for network training, 15% (4 points) for validation, and the remaining 15% (4 points) for network testing (Figure 2C). The output responses were calculated by passing the weighted sum of input variables to each neuron via an activation function represented by the ANN architecture’s hidden layer. The interconnected weights were randomly initialized and adjusted to minimize residual errors between the target and the models’ actual outputs (Figure 2B). The optimal number of neurons in the hidden layer was identified through a systematic trial-and-error method using the TPC input. According to this principle, the best results were acquired with feed-forward network topologies, with three layers: input, output, and one hidden layer, with ten neurons, trained with the back-propagation algorithm. These architectures were then used for all the response variables. The correlation coefficient between experimental response variables and the ANN’s predicted values was higher than 0.9 for training, validation, testing, and overall fitting for all variables. As an example, Figure 2C depicts the scatter plots for TPC modeling. Table 2 shows the high *R^2^* and low RMSE and ADD obtained from the ANN models. *R^2^* values ranged from 0.9802–0.9950. RMSE and ADD values were lower than those of RMS and RSM _ST_, between 0.02–0.24 and 0.19–2.79, respectively. Therefore, it was proved that ANNs are a complex optimization and simulation computational method that displays great potential due to their robust prediction and estimation abilities [34].

### 3.2. Effect of HAE Parameters on the Different Response Variables

RSM regression equations, extraction variables contributions, and statistical parameters (ANOVA) are presented in Table 3. All the response variables adjusted to second-order polynomial equations explained the variation in the different responses as a function of the extraction parameters. The *p*-values were used to evaluate the significance of each coefficient. Low *p*-values, below 0.05, 0.01, and 0.001, indicated that the model terms were significant, highly significant, and remarkably significant, respectively, and *p*-values greater than 0.05 indicate that the model terms were not significant [35]. Temperature (*X_1_*) and S/L ratio (*X_4_*) significantly (*p* < 0.01) contributed to all response variables. Acidity (*X_3_*) significantly (*p* < 0.01) influenced TF, TFL, TPA, TOD, and AC. The impact of time (*X_2_*) was just significant for AC (*p* < 0.05). The quadratic influence of extraction parameters was restricted to TPC (temperature and time), TF (temperature), TFL (temperature and acidity), PAC (S/L ratio), TPA, TOD, and AC (temperature and acidity). Similarly, the interactive effects of the variables were limited to temperature–acidity in TPC; the temperature–S/L ratio in TPC, TFL, PAC, TPA, TOD, and AC; time–acidity in TOD and AC; the time–S/L ratio in TPC and TOD; and acidity–S/L ratio in TFL, PAC, TOD, and AC. The quadratic (5.1–28.6%) and interactive (6.2–17.1%) effects exhibited a low contribution to the models. Contrariwise, the linear effect accounted for 61.3–78.6% of the contribution on the extraction.

The TPC varied from 3.28 to 5.93 mg g^−1^ (Table 1). The model, explaining 94.0% of the variation, was mainly influenced by temperature (35.1%) and the S/L ratio (25.9%) (Table 3). Generally, a high extraction temperature is associated with an increase in the solubility of phenolic compounds from the matrix [36]. A decrease in the S/L ratio enhances the extraction of phenolic compounds from plant matrices by reducing the saturation effects due to the concentration of phenolic compounds [37]. The TF ranged from 5.52–10.10 mg g^−1^ (Table 1). Similar to TPC, temperature exhibited the highest contribution (41.8%), whereas acidity and S/L ratio were the following variables contributing to the variability of the model (15.1% and 12.9%, respectively), which responded to 89.2% of the total variability (Table 3). Previous studies demonstrated a positive impact of this parameter on flavonoid extraction [38]. The TFL oscillated from 0.51 to 1.26 mg g^−1^ (Table 1). The model was mainly contributed by temperature (29.2%), acidity (21.5%), and S/L ratio (13.4%), as observed for TF (Table 3). Recently, Silva et al. [15] observed that higher TFL content was obtained from coffee husk using HAE at 60 °C than with ultrasound-assisted extraction alt 35 °C. Therefore, the extraction temperature showed a key role in the phenolic recovery. The PAC fluctuated less than other responses (2.08–3.48 mg g^−1^) (Table 1). In contrast to other responses, the main factor affecting PAC extraction was the S/L ratio (40.7%) (Table 3). Temperature showed a remarkably significant (*p* < 0.001) effect (27.4%). Procyanidins extraction entails higher difficulty due to the lower polarity in aqueous solvents [39]. As observed, higher water volumes and high temperatures would be needed to recover the maximum PAC yield. The TPA reached 1.19–3.93 mg g^−1^ (Table 1). This extraction model was primarily influenced by temperature and acidity (36.4% and 30.4%, respectively) (Table 3). Likewise, the TOD varied from 0.72 to 1.85 mg g^−1^ (Table 1). Contrary to the other responses, TOD was not only highly influenced by temperature and acidity (24.5 and 21.2, respectively) linearly, but also a remarkably significant (*p* < 0.001) effect was observed quadratically by acidity (20.0%) (Table 3). High acidity can degrade chlorogenic acid. This compound is unstable under these conditions; thus, lower acidity is preferred to extract when chlorogenic acid is present [40]. Finally, the AC of the extracts from the coffee husk oscillated from 9.90 to 18.77 mg g^−1^ (Table 1). Acidity was the main extraction variable affecting AC (42.1%), followed by temperature (28.0%) (Table 3). This model explained 97.4% of the total viability, being the best one among the studied response variables. As previously mentioned, the effect of acidity/low pH needs to be taken into account in extracts with a high concentration of chlorogenic acid, such as coffee, and its by-products [41]. In summary, the positive impact of temperature and S/L ratio was evidenced for all the responses (TPC, TF, TFL, PAC, TPA, TOD, and AC). Temperature increases the water diffusivity, and the lower S/L ratio favors the mass transfer. Moreover, the negative effect of acidity was proved in most of them, attributable to the degradation of chlorogenic acids, highly distributed in all the coffee cherry tissues. Time did not affect the extraction significantly (*p* < 0.05).

All models exhibited remarkably significant fitting (*p* < 0.001), *F*-value (7.07–32.67). *R^2^* values exhibited a very strong correlation, being close to the unity and similar to the Adj. *R^2^* values. To increase the significance of the models’, the non-significant linear, quadratic, and interactive terms were excluded from the model, and the mathematical model was recalculated, resulting in the polynomial equations shown in Table 4. *R^2^* values diminished since these models did not include all the variability of the extraction parameters. Nonetheless, they were much more significant (*p* < 0.0001), showing *F*-values from 15.77 to 44.96. Hence, the obtained models are presented as an approach for predicting the real behavior of the extraction of phenolic compounds from the coffee husk when modifying the studied parameters (temperature, time, acidity, and S/L ratio).

### 3.3. Evaluation and Experimental Validation of Optimal Conditions

Maximizing the desirability of all the responses (TPC, TF, TFL, PAC, TPA, TOD, and AC) conduced to two optimal extraction conditions varying just on the extraction time (100 °C, 0% acid, 0.02 g mL^−1^, and 90 and 5 min, respectively). These conditions predicted the maximum yield of phenolic compounds and antioxidant capacity (among the conditions established). Their responses were evaluated experimentally and validated. 

RSM and ANN-predicted values and experimental results obtained after extraction under optimal and suboptimal conditions are shown in Table 5. The extraction at optimal conditions yielded an extract with a high content of phenolic compounds and antioxidant capacity. The experimental results did not differ from the predicted values from RSM (0.0–5.5%) and ANN (2.3–11.8%). At suboptimal conditions, the difference was slower for the RSM model (0.0–1.6%) and the ANN model (0.2–9.8%). Therefore, both models, showing low CV (%) values, could be validated. The optimal and suboptimal conditions generated extract with significantly different concentrations of phenolic compounds and antioxidant capacity. However, the reduction in time could be of interest to the industry. A considerable time reduction (18-fold) would mean a reduction in energy consumption (the extracting agent, here water, has to reach 100 °C and be maintained for 5 or 90 min). Thus, selecting the suboptimal conditions as the most appropriate for the food industry would result in a higher extraction method sustainability. At these optimal and suboptimal conditions, the need for milling the sample was assessed: extractions were carried using non-milled or raw samples, and the same analyses were performed. No differences (*p* > 0.05) were found for most of the response variables at optimal conditions. On the contrary, all response variables from the raw coffee husk exhibited significant differences (*p* < 0.05) with the milled one. Milling increased the extraction yield by 20–74%. We have previously evidenced that milling increases the extraction of total phenolic compounds from coffee parchment (one of the teguments composing the coffee husk) [42]. A smaller particle size favors mass transfer from the coffee pulp to the extracting water.

Compared with the organic solvent extraction, HAE extraction at optimal and suboptimal conditions yielded 40–70% of TPC, TF, and TOD but extracted 87% of PAC and 100% of TFL and AC. Therefore, while reducing the concentration of phenolic acids, flavanols would be exerting a higher antioxidant capacity, resulting in 100% antioxidant capacity maintenance. The bound phenolic fraction was also studied. This fraction, bound to the coffee husk cell walls, accounted for 11–25% of the total phenolics and antioxidant capacity. From these results, it is evidenced that the residue resulted from the aqueous extraction would still contain a high concentration of free and bound phenolic compounds, and consequently, antioxidant capacity. This residue could be used as a source of antioxidant dietary fiber, as we have recently proposed [43], but also further treated to separate the phenolic compounds associated with dietary fiber [44,45]. The literature gathers scarce information about the extraction of bioactive compounds from the coffee husk. 

The concentration of TPC varies among the diverse extraction conditions used by different authors. Silva et al. [15] extracted phenolic compounds with ethanol and water: ethanol mixtures. Ruesgas-Ramon at al. [46] used deep eutectic solvents, whereas Andrade et al. [47] employed supercritical fluids to extract phenolic compounds from the coffee husk. The aqueous extract presented concentrations similar to those obtained with eutectic solvents but much lower than those obtained with ethanol and supercritical carbon dioxide. Torres-Valenzuela et al. [48] extracted high contents of caffeine, chlorogenic acid, and protocatechuic acid using supramolecular solvents from the coffee pulp. In general, the phytochemical load in the coffee husk depends on the coffee variety, coffee cherry processing, and the plants’ stress and climatic and soil conditions [41,49].

### 3.4. UPLC-ESI-MS/MS Phenolic Compound Profile and Chemometric Analysis

The UPLC-ESI-MS/MS analysis of the phenolic compounds profile from the different coffee husk extracts (Table 6) rendered a better comprehension of the composition of the extracts and the extraction behavior. Representative chromatograms of the optima condition HAE extract, free, and bound phenolic extracts are illustrated in Figure 3A. The main phenolic compounds composing the aqueous extracts was chlorogenic acid (670–906 µg g^−1^), followed by protocatechuic acid (55–128 µg g^−1^), kaempferol-3-*O*-galactoside (12–32 µg g^−1^), and gallic acid (9–23 µg g^−1^). The reduction of the extraction time (from 90 to 5 min) significantly (*p* < 0.05) reduced the concentration of all phenolics compounds, but syringic, *p*-coumaric, and ferulic acids, which were not found or their concentration was reduced in the optimal conditions of extraction (100 °C, 90 min, 0% acid, 0.02 g mL^−1^). Moreover, the effect of milling was also significant (*p* < 0.05). (+)-Catechin and procyanidin B1 were just released from the coffee husk matrix in the milled samples. Likewise, vanillic and 3,4-dihydroxyphenylacetic acids, (−)-epicatechin, and procyanidin B2 were primarily present in the aqueous extract from the optimal conditions and in a much lower concentration when reducing time or skipping the milling step. The free phenolic compounds fraction (extracted with methanol) contained the highest concentration of chlorogenic acid (1428 µg g^−1^), kaempferol-3-*O*-galactoside (40 µg g^−1^), (+)-catechin (30 µg g^−1^), and (−)-epicatechin (25 µg g^−1^). On the other hand, the protocatechuic acid concentration was lower. The high temperature used in HAE may be liberating protocatechuic acid from the bound phenolic fraction [50]. The bound phenolic compounds fraction (extracted after an alkali hydrolysis) was mainly composed of caffeic and protocatechuic acids (90 and 43 µg g^−1^, respectively), with caffeic acid’s concentration being 3.4-fold higher than in the free fraction.

PCA (Figure 3B) revealed the intrinsic grouping among samples. PCA extracted five factors or principal components (PCs) to explain the phytochemical variability among the aqueous and organic extracts from the coffee husk. The two first PCs (the ones graphed) explained 88.1% of the variability; PC1 and PC2 represented 68.7 and 19.4% of the whole variability. The PC1 exhibited a positive influence on all the in vitro determinations (TPC, TF, PAC, TPA, TOD, and AC) but TFL, and most compounds measured by UPLC-MS/MS, excluding caffeic, ferulic, *p*-coumaric, and 3,4-dihydroxyphenylacetic acids, and the flavan-3-ols dimes, procyanidins B1 and B2, which were correlated with PC2. The PCA’s clustering grouped the optimum condition (Op.1 Milled) with the sample of free phenolic compounds. In turn, the three other aqueous extraction conditions (100 °C, 0% acid, 0.02 g mL^−1^; 90 min, using raw coffee pulp, Op.1 Raw; and 5 min using both raw (Op.2 Raw) and milled (Op.2 Milled) coffee pulp) were grouped together, between the free and bound phenolic extracts. Total phenolics were depicted on the right edge of the graph. Therefore, the extracts were classified from left to right according to the total phenolic content.

Figure 3C represents the VIP scores from the PLS analysis. Total phenolics measured by UPLC (total UPLC), phenolic, and hydroxycinnamic acids were the three most variable parameters among the samples. Chlorogenic and caffeic acids were the individual phenolic compounds exhibiting the highest variation, and therefore, showing the most significant impact on sample classification. As the heatmap coupled to the dendrogram of hierarchical clustering shows (Figure 3D), the extract at optimum condition (Op.1) was depic-ted separately, between free and total phenolic, which were considered similar, and bound phenolics and the other three aqueous extracts (Op.1 Raw and Op.2 Milled and Raw). The differences in the extracts’ phenolic composition and antioxidant capacity define the extraction at optimal conditions (100 °C, 90 min, 0% citric acid, and 0.02 g mL^–1^ S/L ratio) as the best extraction, being the most similar to the conventional extraction of free phenolics. A comprehensive analysis was carried out to find 18 compounds, including hydroxybenzoic, hydroxycinnamic, phenylacetic acids, monomeric and dimeric flavan-3-ols, and flavonols. Previous studies have been focused on the main phenolic compounds (chlorogenic, protocatechuic, gallic, and caffeic acids) and the caffeine content [15,46,47].

Phenolic compounds from the coffee husk have been demonstrated to possess antioxidant potential, as revealed in a previous study [51]. Identifying the compounds responsible for these properties arouses great interest. Extensive research has focused on the separation and purification of active biomolecules from food and natural products [52,53]. The isolation and purification of the phytochemicals extracted following the proposed green extraction method could strengthen their biological activity to be then used as food ingredients or nutraceutical products. The ten most significant coefficients from principal component regression (PCR) and partial least squares regression (PLS-R) are depicted in Figure 3E,F. From the PCR coefficients, chlorogenic and protocatechuic acids were the main phenolic compounds responsible for the in vitro antioxidant capacity. According to the PLS-R coefficients, gallic acid and quercetin-3-*O*-glucoside, and 3-*O*-galactoside were also significant contributors to the antioxidant properties of the extracts from the coffee husk. These phenolic compounds have been formerly associated with potent antioxidant properties in vitro and in vivo [54,55,56,57].

Pearson’s correlations were studied to analyze the relationship among in vitro parameters and the phenolic compounds quantified chromatographically. The obtained associations were illustrated in a heatmap (Figure 4).

The concentration of numerous phenolic compounds correlated with the in vitro assays results. The concentration of chlorogenic acid in the extracts from the coffee husk strongly correlated with the content of TPC and TPA (*r* = 0.8973, *p* < 0.01 and *r* = 0.9619, *p* < 0.001, respectively). Protocatechuic acid also showed a strong association with TPA (*r* = 0.8745, *p* < 0.01). Kaempferol-3-*O*-galactoside significantly correlated with TPC (*r* = 0.9008, *p* < 0.001) and TF (*r* = 0.8864, *p* < 0.001). The sum of flavonoids exhibited strong association with TF (*r* = 0.9584, *p* < 0.001). Furthermore, the sum of the concentration of all individual phenolic compounds (total UPLC) presented a significant (*p* < 0.01) correlation (*r* = 0.8333–0.9842) with all the in vitro methods. Consequently, the use of these spectrophotometric techniques to screen the best extraction conditions could be validated, as indicated by Granato et al. [58]. In vitro methods are consistent during screening steps, as long as more specific and comprehensive techniques are used for phytochemical profile analysis.

This study presents the most comprehensive analysis of phenolic compounds composing the coffee husk. Here, we present the aqueous soluble phenolic compounds extracted with HAE and the complete phenolic profile from free and bound phenolic compounds’ fractions. The protective effects of these compounds against oxidative stress and the development of chronic diseases have been widely reported in the literature [51,59]. Chlorogenic acid, the major phenolic compound found in the coffee husk, displayed the highest PCR and PLS-R coefficients, and positively correlated with the in vitro AC (*r* = 0.8977, *p* < 0.01). This compound has been demonstrated to be an excellent radical scavenger following different antioxidant mechanisms and activating cellular antioxidant response (Nrf2-ARE signaling pathways) [54,60]. Additionally, chlorogenic acid presents other biological properties, including the modulations of glucose and lipid metabolism [61], promotion of adipocyte browning [62], and prevention of inflammation [63]. These health-promoting properties elicit the use of the coffee husk as a sustainable source of chlorogenic acid, among other phytochemicals. Thus, using these aqueous extracts from the coffee husk as healthy ingredients could be a great strategy in valorizing coffee by-products and producing novel sustainable products. Although the present work is limited to a variety of coffee husk, the sustainable conditions established could be applied in the extraction of other coffee varieties and even to the extraction of phenolic compounds from the coffee pulp (a by-product comparable to the coffee husk but obtained through the wet processing).

## 4. Conclusions

A green sustainable extraction method for recovering the high-value phenolic compounds from the coffee husk was modeled and validated. The modification of the extraction variables (temperature, time, acidity, and S/L ratio) lead to an improved extraction of phenolic compounds and in vitro antioxidant capacity. The use of RSM and ANN permitted one to model and optimize the aqueous extraction of total phenolic compounds, flavonoids, and flavanols proanthocyanidins, phenolic acids, and *o*-diphenols, and a high in vitro antioxidant capacity. Thus, the optimal conditions (100 °C, 90 min, 0% citric acid, and 0.02 g mL^‒1^ S/L ratio), producing phenolic-rich extracts from the coffee husk using water as the only extracting agent were established. The presence of chlorogenic, protocatechuic, caffeic, and gallic acids and several flavonols, with kaemferol-3-*O*-galactoside being the primary one, was confirmed by the UPLC-ESI-MS/MS results. Chemometric techniques defined chlorogenic acid as the main antioxidant compound. Likewise, multivariate analysis permitted us to validate spectrophotometric techniques for screening the best extraction methods since they showed strong correlations with the chromatographic results. This green extraction may revalorize the coffee husk, a by-product generated globally and of outstanding chemical and biological interest, as a new food ingredient with potential antioxidant and health-promoting properties.

## Figures and Tables

**Figure 1 foods-10-00653-f001:**
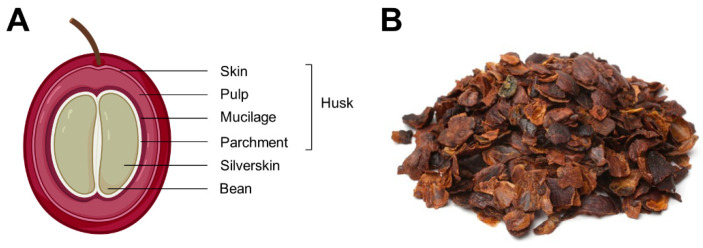
Coffee cherry anatomy (**A**) from the outside to the inside (skin, pulp, mucilage, parchment (comprising the coffee husk), silverskin, and coffee bean) and appearance of dried coffee husk once separated from the coffee bean (**B**).

**Figure 2 foods-10-00653-f002:**
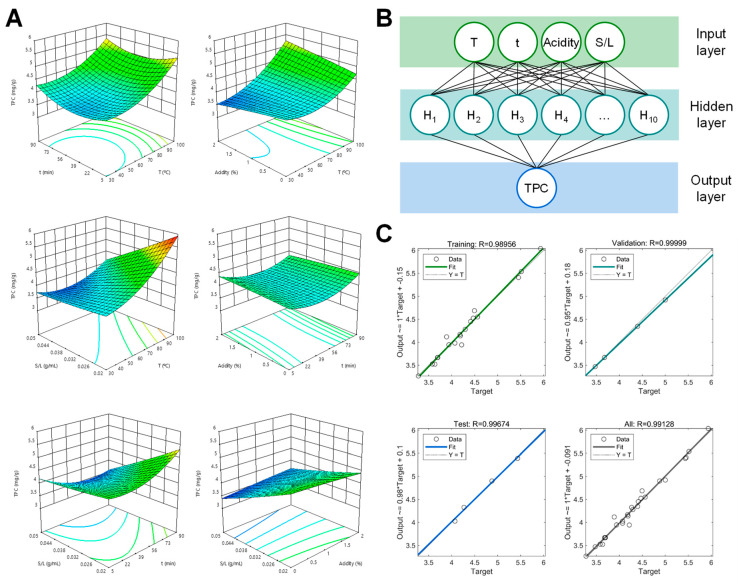
Representative 3D plots illustrating the behavior of total phenolic compounds (TPC) extraction (**A**). Responses (mg g^−1^) are graphed against two paired variables: T (temperature in °C), acidity (% citric acid), t (time in min) and S/L (solid:liquid ratio in g mL^−1^); the topology of the multilayer feed-forward neural network for TPC (**B**), and scatter plot between the experimental and predicted yield by artificial neural networks (ANNs) for training, validation, testing, and overall data fitting for TPC (**C**).

**Figure 3 foods-10-00653-f003:**
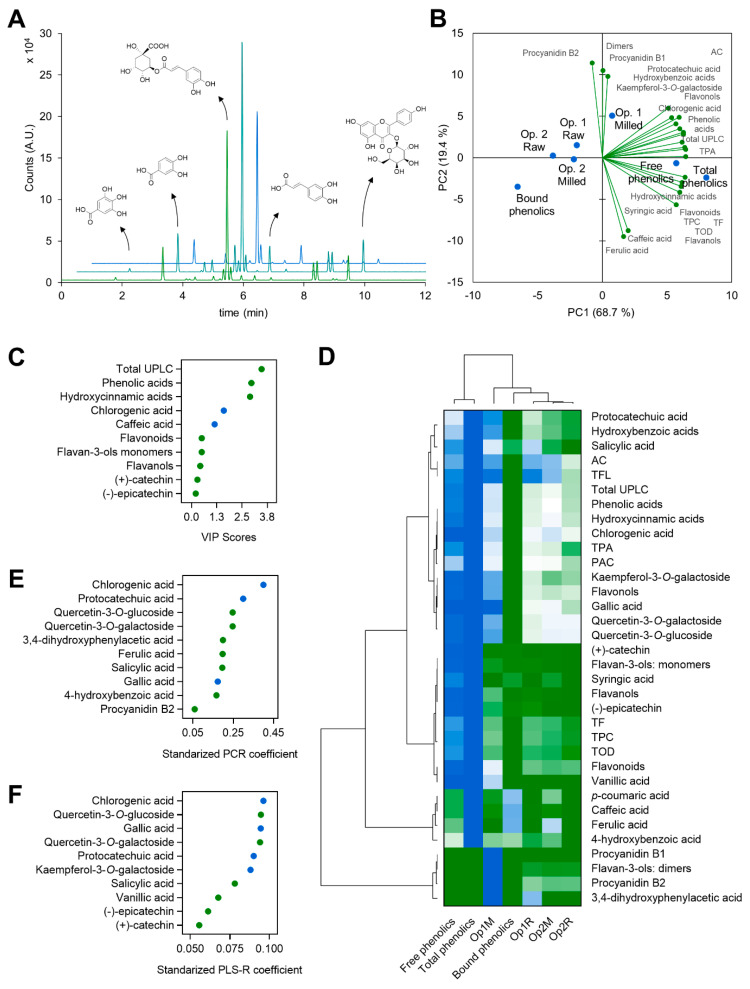
Superimposed chromatograms of the extract at optimal conditions, free, and bound phenolics and the chemical structures of gallic, protocatechuic, chlorogenic, caffeic acids and kaempferol-3-*O*-galactoside, major phenolic compounds found in the coffee husk (**A**), biplot (scores of samples and load factors of each variable) of the principal component analysis (PCA) (**B**), Variable importance in projection (VIP) scores from partial least squares analysis (PLSA) (**C**), agglomerative hierarchical cluster analysis coupled to heatmap (from the lowest (
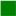
) to the highest (
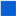
) value for each parameter) (**D**) showing the associations among the measured parameters and classifying phenolic extracts from coffee husk according to them, and the ten most significant coefficients from principal components regression, PCR (**E**) and principal least squares regression, PLS-R (**F**). Circles in different colors indicate minor phenolic or phenolic family, green (
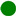
), major phenolic, blue (
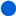
).

**Figure 4 foods-10-00653-f004:**
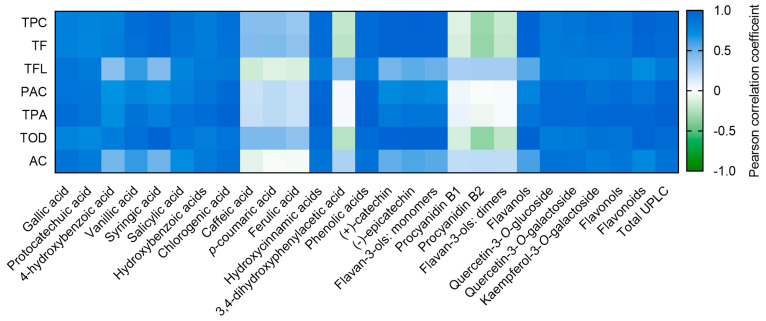
Heatmap depicting the Pearson correlation coefficients from the associations among in vitro determinations of phenolic families and the different phenolic compounds quantified using UPLC-ESI-MS/MS.

**Table 1 foods-10-00653-t001:** Experimental conditions (independent variables) and their corresponding responses values (phenolic compounds content) according to the Box–Behnken design. S/L ratio: solid-to-liquid ratio; TPC: total phenolic compounds; TF: total flavonoids; TFL: total flavanols; PAC: total proanthocyanidin; TPA: total phenolic acid; TOD: total *ortho*-diphenols; AC: antioxidant capacity.

Run	Independent Variables				Responses						
	Temperature (*X_1_*, °C)	Time (*X_2_*, Min)	Acidity (*X_3_*, %)	S/L Ratio (*X_4_*, g mL^−1^)	TPC (mg g^−1^)	TF (mg g^−1^)	TFL (mg g^−1^)	PAC (mg g^−1^)	TPA (mg g^−1^)	TOD (mg g^−1^)	AC (mg g^−1^)
1	30 (−1)	5 (−1)	1 (0)	0.035 (0)	4.19 ± 0.12	5.52 ± 0.27	0.74 ± 0.14	2.48 ± 0.25	1.28 ± 0.08	0.86 ± 0.11	11.95 ± 0.08
2	100 (1)	5 (−1)	1 (0)	0.035 (0)	5.45 ± 0.12	7.94 ± 0.25	0.84 ± 0.07	2.63 ± 0.22	2.29 ± 0.11	1.23 ± 0.06	15.61 ± 0.05
3	30 (−1)	90 (1)	1 (0)	0.035 (0)	4.41 ± 0.09	6.30 ± 0.24	0.73 ± 0.08	2.34 ± 0.09	1.69 ± 0.18	1.03 ± 0.10	11.90 ± 0.09
4	100 (1)	90 (1)	1 (0)	0.035 (0)	5.44 ± 0.10	7.34 ± 0.18	0.96 ± 0.08	2.74 ± 0.13	2.42 ± 0.06	1.28 ± 0.06	15.37 ± 0.07
5	65 (0)	47.5 (0)	0 (−1)	0.02 (1)	4.47 ± 0.13	8.60 ± 0.23	1.09 ± 0.12	2.99 ± 0.27	2.98 ± 0.16	1.56 ± 0.10	17.43 ± 0.05
6	65 (0)	47.5 (0)	2 (1)	0.02 (1)	4.56 ± 0.13	6.38 ± 0.34	0.71 ± 0.03	3.48 ± 0.11	1.55 ± 0.07	1.13 ± 0.05	10.32 ± 0.05
7	65 (0)	47.5 (0)	0 (−1)	0.05 (−1)	3.47 ± 0.15	6.99 ± 0.30	0.75 ± 0.05	2.23 ± 0.26	1.95 ± 0.07	1.10 ± 0.05	13.19 ± 0.05
8	65 (0)	47.5 (0)	2 (1)	0.05 (−1)	4.07 ± 0.10	5.97 ± 0.28	0.70 ± 0.05	2.45 ± 0.30	1.43 ± 0.06	1.00 ± 0.06	11.33 ± 0.09
9	65 (0)	47.5 (0)	1 (0)	0.035 (0)	3.70 ± 0.16	6.81 ± 0.30	0.65 ± 0.09	2.51 ± 0.18	1.77 ± 0.08	0.97 ± 0.05	12.17 ± 0.09
10	30 (−1)	47.5 (0)	1 (0)	0.02 (1)	3.94 ± 0.15	6.69 ± 0.34	0.67 ± 0.05	2.41 ± 0.15	1.37 ± 0.12	1.04 ± 0.06	11.09 ± 0.08
11	100 (1)	47.5 (0)	1 (0)	0.02 (1)	5.93 ± 0.15	10.07 ± 0.24	1.10 ± 0.10	3.04 ± 0.19	2.88 ± 0.10	1.56 ± 0.05	18.67 ± 0.08
12	30 (−1)	47.5 (0)	1 (0)	0.05 (−1)	3.64 ± 0.12	6.17 ± 0.25	0.66 ± 0.09	2.16 ± 0.09	1.69 ± 0.15	0.92 ± 0.08	11.28 ± 0.08
13	100 (1)	47.5 (0)	1 (0)	0.05 (−1)	4.07 ± 0.17	7.32 ± 0.28	0.78 ± 0.06	2.22 ± 0.27	2.19 ± 0.08	1.06 ± 0.05	13.47 ± 0.06
14	65 (0)	5 (−1)	0 (−1)	0.035(0)	4.30 ± 0.21	7.09 ± 0.34	0.76 ± 0.06	2.51 ± 0.20	2.14 ± 0.13	1.26 ± 0.04	16.20 ± 0.08
15	65 (0)	90 (1)	0 (−1)	0.035 (0)	4.27 ± 0.25	7.60 ± 0.27	0.97 ± 0.09	2.55 ± 0.13	2.74 ± 0.10	1.58 ± 0.07	17.33 ± 0.08
16	65 (0)	5 (−1)	2 (1)	0.035 (0)	4.50 ± 0.23	6.89 ± 0.35	0.63 ± 0.14	2.47 ± 0.16	1.54 ± 0.06	1.11 ± 0.06	12.52 ± 0.07
17	65 (0)	90 (1)	2 (1)	0.035 (0)	4.18 ± 0.10	5.55 ± 0.12	0.75 ± 0.09	2.67 ± 0.25	1.53 ± 0.18	1.06 ± 0.07	10.44 ± 0.06
18	65 (0)	47.5 (0)	1 (0)	0.035 (0)	3.68 ± 0.10	6.00 ± 0.21	0.69 ± 0.14	2.52 ± 0.16	1.74 ± 0.11	0.94 ± 0.06	12.04 ± 0.09
19	30 (−1)	47.5 (0)	0 (−1)	0.035 (0)	4.22 ± 0.10	7.06 ± 0.27	0.79 ± 0.10	2.49 ± 0.18	2.23 ± 0.06	1.26 ± 0.07	15.55 ± 0.04
20	100 (1)	47.5 (0)	0 (−1)	0.035 (0)	4.88 ± 0.14	10.10 ± 0.27	1.26 ± 0.09	2.89 ± 0.19	3.93 ± 0.15	1.85 ± 0.07	18.77 ± 0.09
21	30 (−1)	47.5 (0)	2 (1)	0.035 (0)	3.28 ± 0.12	6.54 ± 0.19	0.60 ± 0.06	2.22 ± 0.12	1.28 ± 0.08	0.95 ± 0.04	9.90 ± 0.06
22	100 (1)	47.5 (0)	2 (1)	0.035 (0)	5.00 ± 0.11	8.38 ± 0.21	0.85 ± 0.06	2.88 ± 0.29	2.53 ± 0.08	1.29 ± 0.07	14.12 ± 0.06
23	65 (0)	5 (−1)	1 (0)	0.02 (1)	4.40 ± 0.12	7.47 ± 0.18	0.76 ± 0.10	2.59 ± 0.22	2.09 ± 0.19	1.03 ± 0.04	13.99 ± 0.07
24	65 (0)	90 (1)	1 (0)	0.02 (1)	5.52 ± 0.08	8.15 ± 0.22	0.82 ± 0.09	2.66 ± 0.14	1.89 ± 0.05	1.18 ± 0.05	13.09 ± 0.07
25	65 (0)	5 (−1)	1 (0)	0.05 (−1)	3.89 ± 0.12	7.10 ± 0.18	0.65 ± 0.09	2.25 ± 0.13	1.58 ± 0.07	0.97 ± 0.04	12.76 ± 0.06
26	65 (0)	90 (1)	1 (0)	0.05 (−1)	3.58 ± 0.15	6.67 ± 0.20	0.51 ± 0.14	2.08 ± 0.35	1.19 ± 0.10	0.72 ± 0.04	10.00 ± 0.09
27	65 (0)	47.5 (0)	1 (0)	0.035 (0)	3.69 ± 0.07	6.79 ± 0.17	0.69 ± 0.11	2.59 ± 0.14	1.82 ± 0.07	0.94 ± 0.05	12.45 ± 0.08

**Table 2 foods-10-00653-t002:** Comparison of optimization and prediction capabilities of response surface methodology (RSM) and ANN for the extraction of total phenolic compounds (TPC), total flavonoids (TF), total flavanols (TFL), total proanthocyanidins (PAC), total phenolic acids (TPA), total *o*-diphenols (TOD), and the in vitro antioxidant capacity (AC). *R^2^*: the coefficient of determination; AAD: absolute average deviation.

Response	Modeling Method	*R^2^*	RMSE	AAD (%)
TPC	RSM	0.9402	0.16	3.02
	RSM _ST_	0.9110	0.20	3.58
	ANN	0.9802	0.09	1.48
TF	RSM	0.8919	0.36	4.20
	RSM _ST_	0.7101	0.63	6.95
	ANN	0.9950	0.08	0.57
TFL	RSM	0.9222	0.05	0.50
	RSM _ST_	0.8639	0.06	0.65
	ANN	0.9882	0.02	0.19
PAC	RSM	0.9413	0.06	1.97
	RSM _ST_	0.8747	0.09	2.92
	ANN	0.9879	0.03	0.78
TPA	RSM	0.9355	0.16	6.61
	RSM _ST_	0.8901	0.20	8.40
	ANN	0.9860	0.07	2.79
TOD	RSM	0.9627	0.05	3.47
	RSM _ST_	0.9489	0.06	4.19
	ANN	0.9882	0.03	1.59
AC	RSM	0.9744	0.41	2.47
	RSM _ST_	0.9597	0.51	3.16
	ANN	0.9912	0.24	1.52

RSM _ST_: RSM simplified models including only significant (*p* < 0.05) terms.

**Table 3 foods-10-00653-t003:** Regression coefficient (*β*), contribution, coefficient of determination (*R^2^* and Adj. *R^2^*), and *F*-test value of the predicted second-order polynomial models for the phenolic compounds and antioxidant capacity.

	TPC		TF		TFL		PAC		TPA		TOD		AC	
	*β*	Contrib. (%)	*β*	Contrib. (%)	*β*	Contrib. (%)	*β*	Contrib. (%)	*β*	Contrib. (%)	*β*	Contrib. (%)	*β*	Contrib. (%)
Constant (*X_0_*)	3.689 ***		6.769 ***		0.676 ***		2.541 ***		1.775 ***		0.950 ***		12.221 ***	
Linear		61.3 ***		69.7 ***		65.6 ***		69.5 ***		73.1 ***		61.3 ***		78.6 ***
*X_1_*	0.592 ***	35.1 ***	1.073 ***	41.8 ***	0.133 ***	29.2 ***	0.191 ***	27.4 ***	0.558 ***	36.4 ***	0.185 ***	24.5 ***	2.028 ***	28.0 ***
*X_2_*	0.056	0.3	−0.034	0.0	0.030	1.5	0.010	0.1	0.045	0.2	0.032	0.7	−0.408 *	1.1 *
*X_3_*	−0.001	0.0	−0.645 **	15.1 **	−0.114 ***	21.5 ***	−0.042	1.3	−0.509 ***	30.4 ***	−0.172 ***	21.2 ***	−2.487 ***	42.1 ***
*X_4_*	−0.508 ***	25.9 ***	−0.595 ***	12.9 ***	−0.090 ***	13.4 ***	−0.232 ***	40.7 ***	−0.229 **	6.1 **	−0.144 ***	14.9 ***	−1.045 **	7.4 ***
Quadratic		28.6 ***		8.1 *		17.3 *		5.1		11.5 **		27.2 ***		9.6 ***
*X_1_^2^*	0.559 ***	13.9 ***	0.550 *	4.9 *	0.114 **	9.7 **	0.025	0.2	0.320 **	5.3 **	0.149 ***	7.1 ***	1.255 ***	4.7 ***
*X_2_^2^*	0.517 ***	11.9 ***	−0.179	0.5	0.007	0.0	−0.041	0.6	−0.105	0.6	0.020	0.1	0.439	0.6
*X_3_^2^*	0.153	1.0	0.263	1.1	0.100 **	7.4 **	0.066	1.4	0.324 **	5.5 **	0.251 ***	20.0 ***	1.183* **	4.2 ***
*X_4_^2^*	0.196	1.7	0.317	1.6	0.018	0.2	−0.093 *	2.9 *	−0.055	0.2	0.016	0.1	−0.127	0.0
Interaction		12.5 **		10.9		11.2 *		17.1 **		6.2		9.2 *		10.1 **
*X_12_*	−0.060	0.1	−0.344	1.4	0.032	0.6	0.066	1.1	0.070	0.2	−0.033	0.3	−0.048	0.0
*X_13_*	0.267 *	2.4 *	−0.298	1.1	−0.057	1.8	0.065	1.1	−0.113	0.5	−0.063	1.0	0.249	0.1
*X_14_*	−0.390 **	5.1 **	−0.559	3.8	−0.079 *	3.4 *	−0.145 **	5.3 **	−0.254 *	2.5 *	−0.094 *	2.1 *	−1.345 ***	4.1 ***
*X_23_*	−0.072	0.2	−0.460	2.6	−0.021	0.2	0.042	0.5	−0.153	0.9	−0.089 *	1.9 *	−0.799 *	1.4 *
*X_24_*	−0.357 *	4.3 *	−0.280	1.0	−0.051	1.4	−0.060	0.9	−0.047	0.1	−0.099 *	2.3 *	−0.465	0.5
*X_34_*	0.126	0.5	0.301	1.1	0.082*	3.7 *	0.182 **	8.3 **	0.228	2.0	0.082 *	1.6 *	1.312 **	3.9 **
Model		94.0 ***		89.2 ***		92.2 ***		94.1 ***		93.6 ***		96.3 ***		97.4 ***
*R^2^*	0.9402		0.8919		0.9222		0.9413		0.9355		0.9627		0.9744	
Adj. *R^2^*	0.8705		0.8296		0.8314		0.8728		0.8603		0.9192		0.9446	
*F* value (model)	13.48 ***		7.07 ***		10.16 ***		13.75 ***		12.43 ***		22.13 ***		32.67 ***	

*X_1_*: extraction temperature (°C), *X_2_*: extraction time (min), *X_3_*: acidity (% citric acid), *X_4_*: solid-to-liquid ratio (g mL^−1^), *R*^2^: Coefficient of determination. Level of significance: * *p* < 0.05, ** *p* < 0.01, *** *p* < 0.001.

**Table 4 foods-10-00653-t004:** Non-coded equations and their statistical parameters for the extraction of total phenolic compounds (TPC), total flavonoids (TF), total flavanols (TFL), total proanthocyanidins (PAC), total phenolic acids (TPA), total *o*-diphenols (TOD), and the in vitro antioxidant capacity (AC).

Non-Coded Equation	*R^2^*	*F*-Value	*p*-Value
*Y_TPC_* = 4.0 − 1.5 × 10^−2^ *x_1_* − 1.7 × 10^−3^*x_2_* − 5.0 × 10^−1^*x_3_* + 4.1 × 10^+1^*x_4_* + 3.8 × 10^−4^*x_1_^2^* + 2.4 × 10^−4^*x_2_^2^* + 7.6 × 10^−3^*x_13_* − 7.4 × 10^−1^*x_14_* − 5.6 × 10^−1^*x_24_*	0.9110	19.33	<0.0001
*Y_TF_* = 8.6 − 1.9 × 10^−2^*x_1_* − 6.4× 10^−1^ *x_3_* − 4.0× 10^+1^ *x_4_* + 3.8× 10^−4^ *x_1_^2^*	0.7414	15.77	<0.0001
*Y_TFL_* = 1.1 − 2.4 × 10^−3^*x_1_* − 4.9 × 10^−1^*x_3_* − 1.7*x_4_* + 8.8 × 10^−5^*x_1_^2^* + 9.4 × 10^−2^*x_3_^2^* − 1.5 × 10^−1^*x_14_* + 5.4*x_34_*	0.8639	17.23	<0.0001
*Y_PAC_* = 2.0 + 1.5 × 10^−2^*x_1_* − 4.7 × 10^−1^*x_3_* + 2.2 × 10^+1^*x_4_* − 4.6 × 10^+2^*x_4_^2^* − 2.8 × 10^−1^*x_14_* + 1.2 × 10^+1^*x_34_*	0.8747	23.27	<0.0001
*Y_TPA_* = 2.2 − 5.3 × 10^−3^*x_1_* − 1.2*x_3_* + 1.6 × 10^+1^*x_4_* + 2.9 × 10^−4^*x_1_^2^* + 3.6 × 10^−1^*x_3_^2^* − 4.8 × 10^−1^*x_14_*	0.8901	26.99	<0.0001
*Y_TOD_* = 1.3 − 3.3 × 10^−3^*x_1_* + 8.3 × 10^−3^*x_2_* − 7.5 × 10^−1^*x_3_* + 4.0*x_4_* + 1.1 × 10^−4^*x_1_^2^* + 2.4 × 10^−1^*x_3_^2^* − 1.8 × 10^−1^*x_14_* − 2.1 × 10^−3^*x_23_* − 1.6 × 10^−1^*x_24_* + 5.4*x_34_*	0.9489	29.73	<0.0001
*Y_AC_* = 1.6 × 10^+1^ + 2.3 × 10^−2^*x_1_* + 9.2 × 10^−3^*x_2_* − 6.9*x_3_* + 9.4*x_4_* + 9.6 × 10^−4^*x_1_^2^* + 1.1*x_3_^2^* − 2.6*x_14_* − 1.9 × 10^−2^*x_23_* + 8.7 × 10^+1^*x_34_*	0.9597	44.96	<0.0001

**Table 5 foods-10-00653-t005:** Validation of predicted values at optimal conditions of the aqueous extraction and comparison with organic solvent extraction of the phenolic compounds from the coffee husk.

Response	Optimal Conditions Aqueous Extraction								Organic Solvent Extraction		
	100 °C, 90 min, 0% acid, 0.02 g mL^−1^				100 °C, 5 min, 0% acid, 0.02 g mL^−1^				MeOH:H_2_O	NaOH-AcEt	Ʃ
	Predicted (CV, %)		Experimental		Predicted (CV, %)		Experimental		*Free* *Phenolics*	*Bound* *Phenolics*	*Total* *Phenolics*
	RSM	ANN	Milled	Raw	RSM	ANN	Milled	Raw			
TPC (mg g^−1^)	6.56 (3.5)	5.83 (11.8)	6.89 ± 0.13 ^d^	6.31 ± 0.46 ^d^	5.70 (1.4)	4.97 (8.2)	5.59 ± 0.06 ^c^	3.94 ± 0.13 ^b^	15.99 ± 1.16 ^e^	2.39 ± 0.29 ^a^	18.38
TF (mg g^−1^)	11.29 (1.2)	9.82 (11.0)	11.48 ± 0.05 ^d^	11.01 ± 0.41 ^d^	10.56 (0.8)	9.43 (7.2)	10.44 ± 0.08 ^c^	7.08 ± 0.05 ^b^	26.82 ± 2.65 ^e^	4.72 ± 0.47 ^a^	31.55
TFL (mg g^−1^)	1.60 (5.5)	1.61 (5.1)	1.73 ± 0.07 ^d^	1.67 ± 0.06 ^d^	1.33 (1.6)	1.18 (9.8)	1.36 ± 0.04 ^c^	0.78 ± 0.01 ^b^	1.63 ± 0.30 ^d^	0.20 ± 0.06 ^a^	1.83
PAC (mg g^−1^)	3.42 (0.0)	3.14 (5.9)	3.42 ± 0.05 ^d^	3.24 ± 0.09 ^c^	3.23 (0.2)	3.06 (3.7)	3.22 ± 0.12 ^c^	2.68 ± 0.03 ^b^	3.91 ± 0.37 ^e^	1.32 ± 0.08 ^a^	5.23
TPA (mg g^−1^)	4.87 (0.5)	5.01 (2.5)	4.84 ± 0.48 ^d^	4.12 ± 0.29 ^c^	3.90 (0.8)	3.76 (3.4)	3.94 ± 0.18 ^c^	23.64 ± 0.08 ^b^	6.88 ± 0.23 ^e^	0.98 ± 0.06 ^a^	7.86
TOD (mg g^−1^)	2.25 (1.6)	1.97 (7.9)	2.20 ± 0.06 ^e^	2.04 ± 0.03 ^d^	1.87 (0.0)	1.88 (0.2)	1.87 ± 0.08 ^c^	1.28 ± 0.06 ^b^	5.56 ± 0.30 ^f^	0.91 ± 0.11 ^a^	6.47
AC (mg g^−1^)	23.64 (0.7)	22.69 (2.3)	23.42 ± 0.15 ^e^	22.36 ± 0.24 ^d^	21.12 (0.8)	19.35 (7.0)	21.36 ± 0.66 ^d^	14.99 ± 0.18 ^b^	22.79 ± 1.54 ^d e^	5.31 ± 0.42 ^a^	28.10

Results are reported as mean ± SD (*n* = 3). Mean values followed by different superscript letters significantly differ (among columns) when subjected to ANOVA analysis and Tukey multiple range *post hoc* test (*p* < 0.05).

**Table 6 foods-10-00653-t006:** UPLC-ESI-MS/MS phenolic compounds profile of the coffee husk extracts obtained by the aqueous extraction using optimal conditions and the organic solvent extraction of the free and bound phenolic fractions.

Compound (µg g^−1^)	*R_t_* (min)	Mass Spectral Data		Optimal Conditions Aqueous Extraction				Organic Solvent Extraction		
		[M − H]^−^ (*m/z*)	MS^2^ (*m/z*)	100 °C, 0% acid, 0.02 g mL^−1^				MeOH:H_2_O	NaOH-AcEt	Ʃ
				90 Min		5 Min		*Free* *Phenolics*	*Bound* *Phenolics*	*Total* *Phenolics*
				Milled	Raw	Milled	Raw			
*Hydroxybenzoic acids*										
Gallic acid	1.73	169	125	22.79 ± 1.45 ^d^	10.94 ± 0.81 ^b^	12.11 ± 0.89 ^c^	8.51 ± 0.10 ^a^	23.15 ± 0.30 ^d^	-	23.15
Protocatechuic acid	3.34	153	109	127.96 ± 6.86 ^f^	83.53 ± 5.37 ^d^	66.20 ± 8.73 ^c^	55.08 ± 2.11 ^b^	99.94 ± 1.33 ^e^	42.82 ± 8.59 ^a^	142.76
4-hydroxybenzoic acid	4.43	137	93	3.51 ± 0.34 ^c^	2.47 ± 0.19 ^b^	3.19 ± 0.36 ^c^	1.56 ± 0.07 ^a^	4.30 ± 0.07 ^d^	3.75 ± 0.51 ^c d^	8.06
Vanillic acid	5.43	167	152	6.00 ± 2.52 ^a^	-	-	-	9.55 ± 0.20 ^b^	-	9.55
Syringic acid	5.96	197	182	-	-	0.20 ± 0.03 ^a^	-	1.66 ± 0.10 ^b^	0.20 ± 0.01 ^a^	1.86
Salicylic acid	8.96	137	93	0.81 ± 0.03 ^b^	0.90 ± 0.08 ^b^	0.22 ± 0.04 ^a^	-	1.19 ± 0.02 ^c^	0.25 ± 0.03 ^a^	1.43
*Hydroxycinnamic acids*										
Chlorogenic acid	5.38	353	191	905.67 ± 18.50 ^e^	747.17 ± 36.10 ^c^	840.04 ± 30.79 ^d^	669.54 ± 40.06 ^b^	1428.40 ± 25.80 ^f^	0.67 ± 0.08 ^a^	1429.07
Caffeic acid	5.48	179	135	15.16 ± 0.51 ^b^	10.51 ± 1.07 ^a^	14.00 ± 1.50 ^b^	9.35 ± 0.11 ^a^	26.21 ± 0.79 ^c^	89.93 ± 9.73 ^d^	116.14
*p*-coumaric acid	6.81	163	119	2.27 ± 0.04 ^b^	1.39 ± 0.11 ^a^	4.22 ± 0.14 ^d^	1.37 ± 0.06 ^a^	2.94 ± 0.04 ^c^	8.21 ± 1.31 ^e^	11.15
Ferulic acid	7.81	193	134	-	-	4.25 ± 0.36 ^b^	-	1.74 ± 0.08 ^a^	5.08 ± 0.67 ^c^	6.81
*Phenylacetic acids*										
3,4-dihydroxyphenylacetic acid	4.18	167	123	1.46 ± 0.51 ^b^	1.04 ± 0.04 ^a^	-	-	-	-	-
*Flavan-3-ols: monomers*										
(+)-catechin	5.22	289	245	0.45 ± 0.05 ^b^	-	0.31 ± 0.03 ^a^	-	30.31 ± 0.28 ^d^	0.58 ± 0.08 ^c^	30.89
(−)-epicatechin	6.27	289	245	4.71 ± 0.51 ^c^	1.45 ± 0.06 ^b^	-	-	25.07 ± 0.23 ^d^	0.64 ± 0.05 ^a^	25.71
*Flavan-3-ols: dimers*										
Procyanidin B1	4.90	577	289	5.84 ± 0.69	-	-	-	-	-	-
Procyanidin B2	5.93	577	289	3.03 ± 0.46 ^c^	0.93 ± 0.17 ^b^	0.76 ± 0.09 ^a^	0.77 ± 0.08 ^a^	-	-	-
*Flavonols*										
Quercetin-3-*O*-glucoside	8.34	463	301	15.03 ± 0.96 ^d^	8.89 ± 1.17 ^b^	10.45 ± 0.73 ^c^	10.39 ± 0.44 ^c^	18.16 ± 0.42 ^e^	0.66 ± 0.03 ^a^	18.82
Quercetin-3-*O*-galactoside	8.65	463	301	14.33 ± 0.14 ^d^	8.79 ± 0.42 ^b^	10.20 ± 1.91 ^c^	10.25 ± 0.26 ^c^	18.17 ± 0.41 ^e^	0.67 ± 0.05 ^a^	18.84
Kaempferol-3-*O*-galactoside	9.46	447	284	32.12 ± 0.94 ^e^	18.27 ± 0.83 ^d^	11.50 ± 0.24 ^b^	14.13 ± 1.46 ^c^	40.32 ± 0.85 ^f^	1.05 ± 0.11 ^a^	41.37

Results are reported as mean ± SD (*n* = 3). Mean values followed by different superscript letters significantly differ (among columns) when subjected to ANOVA analysis and Tukey’s multiple range *post hoc* test (*p* < 0.05).

## Data Availability

The data presented in this study are available from the corresponding author on reasonable request.

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
