# Peer review of "Revalorization of Coffee Husk: Modeling and Optimizing the Green Sustainable Extraction of Phenolic Compounds"

_foods, 2021, doi:10.3390/foods10030653_

Round 1

Reviewer 1 Report

What an outstanding and well-written piece of research! This is a truly comprehensive, well-designed and executed study that provides a wealth of information on the factors that may affect extraction of phenolics from coffee husks. The RSM and AI modeling approaches are perfect for the purposes of the study, and the results are impressive in terms of goodness-of-fit of the various models.

Great analytical chemistry, with state-of-the-art techniques.

The manuscript is long, with many tables and figures, but frankly, the text and all the supporting materials are all very much needed.

The study is well positioned in the context of the existing literature, but in the introduction and in the discussion.

The conclusions are well warranted.

The only matter I would suggest raising in the discussion as a limitation of the study is that it was performed on one single coffee. It would indeed have been great to validate your findings across several coffees (in terms of variety and origins), maybe.

I don't often recommend publication as is, but this is one of those instances. Kudos and thanks (o enhorabuena y gracias)!

Author Response

Reviewer 1

The only matter I would suggest raising in the discussion as a limitation of the study is that it was performed on one single coffee. It would indeed have been great to validate your findings across several coffees (in terms of variety and origins), maybe.

  • We sincerely thank the reviewer comments. Although we did not evaluate the usefulness of the extraction method in other coffee varieties. We are currently using it to produce coffee pulp extracts, which have been used in different research, just published, to date, as proceeding. The method seemed to be robust and leads to similar extracts, just varying according to the initial chemical composition of the coffee by-product.

Braojos, C.; Rebollo-Hernanz, M.; Benitez, V.; Cañas, S.; Aguilera, Y.; Arribas, S.M.; Martin-Cabrejas, M.A. Simulated Gastrointestinal Digestion Influences the In Vitro Hypolipidemic Properties of Coffee Pulp, a Potential Ingredient for the Prevention of Non-Alcoholic Fatty Liver Disease. Proceedings 202061, 19.

https://doi.org/10.3390/IECN2020-06997

  • We have included a brief paragraph indicating the limitation of this study.
  • Paragraph added (page 19, line 589): Although the presents work is limited to a variety of coffee husk, the sustainable conditions established could be applied in the extraction of other coffee varieties and even to the extraction of phenolic compounds from the coffee pulp (a by-product comparable to the coffee husk but obtained through the wet processing).

    Finally, we do want to thank the reviewer’s comments for his/her time and meticulous care in reviewing our manuscript.

Reviewer 2 Report

The authors research the coffee husk (cascara), which is produced in large amounts during coffee processing, to extract chlorogenic acid and other polyphenols. For that, the authors optimized a simple water extraction method. They have also characterised the polyphenolic fraction using UPLC-MS/MS analysis. The modern, ecologically oriented society attaches great importance to waste reduction, so it makes sense not to dispose of the by-products of coffee production and to bring them into the value chain. An added value of the coffee plant could increase social and economic prosperity in poorer coffee-growing regions and work against the decreasing coffee price, which is especially worthwhile in the current times of a global economic crisis.

The research methods are described in a fashion allowing reproducibility, and the conclusions are based in the available data.

I have only some minor suggestions:

Around line 50: it could be mentioned that cascara is already applied in some countries such as Yemen for preparation of a traditional beverage.

Line 92: could you specify the variety of Arabica used. More details on processing would also be appreciated.

Line 115: what was the rationale to select a second order polynomial already at this stage. Typically, with Design Expert software, you have the option to select the model based on the best-fit to the data (and even select different models such as linear or polynomial dependent on compound).

Table 1: add units in section independent variables

Figure 2: the resolution is very low and the small print difficult to read

Section 2.2.2.: regarding the appropriateness of ANN, I wonder if not the data amount is quite low for such a method, which could lead to various overfitting problems leading to models with high R2 which will nevertheless be not very robust to new data (e.g. how can the validation R be higher than the training R in Figure 2?). Is ANN really providing better insight into the data than the conventional Design Expert ANOVA model? Currently, I would tend to suggest the deletion of the ANN part of the paper.

Table 4: this could be moved to an annex

Line 430: have should read has.

References: check journal style. DOI information is missing?

Author Response

Reviewer 2

Around line 50: it could be mentioned that cascara is already applied in some countries such as Yemen for preparation of a traditional beverage.

  • We have included the traditional usage of the coffee husk to produce the cascara beverage in the text.
  • Paragraph added (page 2, line 50): The traditional use of the coffee husk is the preparation of the “Cascara beverage”, traditionally consumed in Yemen and Ethiopia [7].
  • Reference added:
  1. Heeger, A.; KosiÅ„ska-Cagnazzo, A.; Cantergiani, E.; Andlauer, W. Bioactives of coffee cherry pulp and its utilisation for production of Cascara beverage. Food Chem. 2017, 221, 969–975, doi:10.1016/j.foodchem.2016.11.067.

Line 92: could you specify the variety of Arabica used. More details on processing would also be appreciated.

  • The variety of the coffee husk has been added (Caturra) in the text as well as the main steps for obtaining the coffee husk from the coffee cherry.
  • Paragraph modified (page 3, line 94): The coffee husk, mechanically separated from the sun-dried cherries of the Arabica species variety Caturra, was supplied by “Las Morenitas” (Nicaragua).

Line 115: what was the rationale to select a second order polynomial already at this stage. Typically, with Design Expert software, you have the option to select the model based on the best-fit to the data (and even select different models such as linear or polynomial dependent on compound).

  • We agree with the reviewer comment. When running the experiment on the Design Expert software we did not chose any model, however, when analyzing the results, the software suggests the best fitting option, according to the F-and p-values for the fit and for the lack of fit of each model (linear, 2FI, quadratic, cubic). Thus, we just selected quadratic as the statistical program recommended it.

Table 1: add units in section independent variables

  • The units have been added in the table according to the reviewer’s suggestion.

Figure 2: the resolution is very low and the small print difficult to read

  • Figure 2 have been changed for a high-quality figure with enhanced resolution in which numbers and words are readable.

Section 2.2.2.: regarding the appropriateness of ANN, I wonder if not the data amount is quite low for such a method, which could lead to various overfitting problems leading to models with high R2 which will nevertheless be not very robust to new data (e.g. how can the validation R be higher than the training R in Figure 2?). Is ANN really providing better insight into the data than the conventional Design Expert ANOVA model? Currently, I would tend to suggest the deletion of the ANN part of the paper.

  • The present study bases both RSM and ANN models on a data set composed of 27 experiments, constructed based on a Box-Behnken experimental design. Previous reports have used a similar number of conditions or even less (18). Evidence demonstrates that ANN generated good models that lose accuracy with extreme values. In the present study, the model was validated using the optimal conditions for extraction, which tended to choose the highest temperatures and times, and lowest acidity and solid-to-liquid ratios. Therefore, the model can be slightly biased due to the characteristics of the experimental design (Box-Behnken) which concentrates most of the conditions in the middle level. That can, nonetheless, also happen using RSM.
  • Carbone, K.; Amoriello, T.; Iadecola, R. Exploitation of Kiwi Juice Pomace for the Recovery of Natural Antioxidants through Microwave-Assisted Extraction. Agriculture202010, 435. https://doi.org/10.3390/agriculture10100435
  • Li, C.; Cui, Y.; Lu, J.; Liu, C.; Chen, S.; Ma, C.; Liu, Z.; Wang, J.; Kang, W. Ionic Liquid-Based Ultrasonic-Assisted Extraction Coupled with HPLC and Artificial Neural Network Analysis for Ganoderma lucidumMolecules202025, 1309. https://doi.org/10.3390/molecules25061309
  • Osman, H.; Shigidi, I.; Arabi, A. Multiple Modeling Techniques for Assessing Sesame Oil Extraction under Various Operating Conditions and Solvents. Foods20198, 142. https://doi.org/10.3390/foods8040142
  • The R2 values for validation, are independent to those of training. The created model (in training) is used in validation. If the model has a constant tendency to underestimate the content of phenolics (TPC in Figure 2), but this tendency is constant, R2 values will be higher. However, the actual values will be higher. This happened during mathematical validation (Figure 2) and during the experimental validation.
  • Nevertheless, the authors include both methods to compare the modeling characteristics and optimization utility of both methods. Hence, we consider that ANN, although not giving a better accuracy (just similar) than RSM, should be maintained in the manuscript.

Table 4: this could be moved to an annex

  • The authors have considered to maintain the table where it is currently positioned. Since it is a small table which does not disturb the reading, we consider that it will be better to maintain its present location, thus helping the reader to find the results better.

Line 430: have should read has.

  • The wording has been changed according to the suggestion.

References: check journal style. DOI information is missing?

  • All references have been checked and the DOI information has been included in all cases.

Finally, we do want to thank the reviewer’s comments for his/her time and meticulous care in reviewing our manuscript.